



# Peroxy Radical Chemistry and the Volatility Basis Set

Meredith Schervish[1] and Neil M. Donahue[1]

[1]Center for Atmospheric Particle Studies, Carnegie Mellon University, 5000 Forbes Avenue, Pittsburgh PA 15213 USA

**Correspondence:** Neil M. Donahue (nmd@andrew.cmu.edu)

**Abstract.** Gas-phase auto-oxidation of organics can generate highly-oxygenated organic molecules (HOMs) and thus increase secondary organic aerosol production and enable new-particle formation. Here we present a new implementation of the Volatility Basis Set (VBS) that explicitly resolves peroxy radicals ($RO_2$) formed via auto-oxidation. The model includes a strong temperature dependence for auto oxidation as well as explicit termination of $RO_2$, including reactions with NO, $HO_2$, and other $RO_2$. The $RO_2$ cross reactions can produce dimers (ROOR). We explore the temperature and $NO_x$ dependence of this chemistry, showing that temperature strongly influences the intrinsic volatility distribution and that NO can suppress auto-oxidation under conditions typically found in the atmosphere.

## 1 Introduction

Atmospheric particles have dangerous health effects and influence the environment both through their direct scattering of incoming radiation and their indirect effect on clouds as cloud condensation nuclei. New-particle formation (secondary particle production) constitutes the largest source of particles in the atmosphere; furthermore many if not most particles with significant health or climate effects consist largely of secondary material, having grown via condensation from a much smaller initial size. Organic vapors contribute to both nucleation and growth of new particles as well as growth of primary nano particles, all of which constitutes secondary organic aerosol (SOA). Especially for condensation to the smallest particles, the volatility of those condensible organics is important. Therefore it is of interest to investigate and model how volatile precursors oxidize in the atmosphere to low volatility or extremely-low volatility products (LVOCs and ELVOCs) that will contribute to SOA. Furthermore, organic oxidation products alone can drive "pure biogenic" nucleation (Kirkby et al., 2016), and so the formation mechanisms and nucleation rates of a new class of ultra-low volatility products (ULVOCs) that can drive nucleation is also of interest.

One important class of condensible organics is the group known as highly-oxygenated organic molecules (HOMs). HOMs are are organic compounds containing at least six oxygen atoms formed through peroxy-radical ($RO_2$) isomerization and subsequent addition of molecular oxygen, which preserves $RO_2$ functionality in a recursive process known as auto-oxidation, followed by $RO_2$ termination to molecular products (Crounse et al., 2013; Ehn et al., 2014; Bianchi et al., 2019). HOMs open up a rich space for gas-phase organic oxidation chemistry to strongly influence new-particle formation involving condensible organic products. Because the chemistry involves competition between internal isomerization (which typically will have a relatively high energy barrier and thus a strong temperature dependence) and radical-radical termination reactions (which





typically have low barriers and thus only a modest temperature dependence), we expect a strong temperature dependence to HOM formation. Furthermore, because $NO_x$ (especially NO) can greatly shorten $RO_2$ lifetimes, we also expect a strong $NO_x$ dependence to HOM formation. Finally, to the extent that $RO_2$-$RO_2$ cross reactions between radicals from different organic precursors can be important, there may be a rich interplay among the oxidation mechanisms for a host of different organic

compounds.

The volatility basis set (VBS) has been developed and used widely as a framework to represent, track, and model the oxidation of volatile organic precursors, the subsequent production of condensible products, and their evolution through multiple generations of oxidation chemistry and fragmentation (Donahue et al., 2005, 2006, 2011, 2012b, 2013; Chuang and Donahue, 2016). For numerical calculations, the 2D-VBS space is discretized into bins, with each $C^*$(300) bin typically separated by 1

order of magnitude and each O:C bin separated by 0.1. It is not necessary to know the identity of molecules in these bins, but this can be inferred by application of composition activity relations such as SIMPOL (Pankow and Asher, 2007). In general, the perspective of the VBS has been to explicitly track the most important observable properties of organics with respect to organic-aerosol formation – volatility ($C^\circ$) and the degree of oxidation (O:C) – forming the so-called two-dimensional volatility basis set (2D-VBS). Where specific products have known properties, those molecules can be located individually in the

2D-VBS. Where specific products are not known, the 2D-VBS can still track the overall carbon in a reactive system, with properties ($C^\circ$ and O:C) evolving following known chemistry during photochemical aging.

The importance of $RO_2$ branching chemistry has so far been recognized implicitly in VBS formulations (Presto and Donahue, 2006; Henry and Donahue, 2011), but not explicitly. Instead, the product (volatility) distribution in the VBS has been based on some measure of the $RO_2$ branching (i.e. VOC:$NO_x$ or $RO_2$ branching in an explicit mechanism). Consequently, there

have been "high-$NO_x$" products and "low-$NO_x$" products (volatility distributions) (Presto and Donahue, 2006), or even "high-$HO_2$" and "high-$RO_2$" products (Henry and Donahue, 2011), that can be mixed to form products at intermediate $NO_x$ (or $HO_x$:$RO_x$) based on this indirect measure of $RO_2$ branching. Further, relatively little has been done to represent any temperature dependence of VOC oxidation in VBS implementations.

The purpose of this paper is to extend the 2D-VBS to explicitly represent $RO_2$ chemistry, including $RO_2$ oxidation and

HOM formation as well as all forms of $RO_2$ termination, and only distribute products into volatility bins after $RO_2$ termination in an explicit gas-phase chemical mechanism. In this way we can represent the gas-phase HOM formation mechanism, its temperature dependence, and the rich array of competing $RO_2$ termination processes in a scheme that can merge seamlessly with any gas-phase chemical mechanism (MCM, etc.) while maintaining the generality of the VBS and its ability to represent the ongoing chemistry of organic species associated with organic aerosol formation and particle growth.

## 2   Background

HOMs and their properties have been measured in multiple laboratory and field studies. HOMs can drive pure biogenic nucleation indicating that, in the VBS context, at least some of them must be ELVOCs/ULVOCs. Stolzenburg *et al.* (Stolzenburg et al., 2018) and Tröstl *et al.* (Tröstl et al., 2016) report measurements of HOMs in the LVOC and SVOC range during ex-





periments in the CLOUD chamber at CERN, showing that these HOMs (especially LVOCs) drive growth during new-particle formation, providing a significant reservoir of condensible species to stabilize particles after initial clustering (Stolzenburg et al., 2018; Tröstl et al., 2016). From these data, both collected under $NO_x$-free conditions, HOM products are expected to be distributed in SVOC, LVOC, and ELVOC range, with the largest concentration as LVOCs.

The composition and properties of HOMs produced during oxidation of a precursor such as $\alpha$-pinene (as well as their quantitative yields) depend on the conditions under which they are formed. Frege *et al.*, again at CLOUD, found that at lower temperatures fewer total HOMs are formed, with dimers seeing the sharpest fall-off (Frege et al., 2018). Lehtipalo *et al.* showed that increasing $NO_x$ concentrations suppress the ELVOC HOMs but have less of an effect on the LVOC and SVOC HOMs (Lehtipalo et al., 2018).

There are two important elements to HOM formation and their volatility distribution. The first is auto-oxidation itself, which produces progressively more oxidized peroxy radicals via H-atom transfer and thus -OOH functionalization before the radicals ultimately terminate to form stable products. The second is dimerization, which is one of the termination processes, where these functionalized $RO_2$ associate to form a covalently bonded dimer (presumably a peroxide, ROOR). The first process plays a key role in particle growth and the temperature dependence of condensible vapor yields. The second process may be rate

limiting for pure biogenic nucleation.

Auto-oxidation involves an internal H-atom transfer, which almost certainly has a significant activation energy. Further, the process is observed to be competitive with bimolecular reactions on a timescale of 1 s or longer (Ehn et al., 2014). Unimolecular reactions have an intrinsic rate constant (A-factor) given by molecular vibrational frequencies, and even loose bending modes in molecules have frequencies of THz. To slow down unimolecular reactions to 1 $s^{-1}$ or so, a high activation energy is thus

almost essential. Rate coefficients for competitive reactions suggest an auto-oxidation energy barrier in the range of 15-18 kcal $mol^{-1}$ (7500-9000 K). This is also consistent with quantum chemical calculations as well as experimental data fitted to an extended MCM model (Rissanen et al., 2014; Molteni et al., 2019). This high activation energy also means that the $RO_2$ isomerization reactions will have a strong temperature dependence, becoming very slow at low temperature.

In contrast to the strong temperature dependence of auto oxidation, the two canonical $RO_2$ loss reactions, with $HO_2$ and NO,

both have only a weak temperature dependence with similar rate coefficients near $10^{-11}$ $cm^3$ $molec^{-1}$ $s^{-1}$ largely independent of $RO_2$ structure (Atkinson et al., 2008). $HO_2$ levels typical in the lower atmosphere are roughly 3 pptv, or $10^8$ molec $cm^{-3}$, giving a first-order low-$NO_x$ loss frequency of $10^{-3}$ $s^{-1}$. NO in polluted areas can go much higher, increasing the high-$NO_x$ loss to 0.1 $s^{-1}$ or even higher. This establishes the competitive range for $RO_2$ first-order loss in the 0.001 - 1 $s^{-1}$ range in the atmosphere. Because these rate coefficients are only weakly sensitive to temperature, the high activation energy for the H-atom

transfer in auto-oxidation means we expect the reactions with $HO_2$ and NO to be more competitive at low temperature.

Another $RO_2$ loss is to react with other $RO_2$. Unlike their reactions with $HO_2$ and NO, in which the rate coefficients depend only weakly on the nature of the R group, the rate coefficients for $RO_2$-$RO_2$ self and cross reactions are extremely sensitive to the nature of the R group. This has been well established for decades for the self reactions of different small $RO_2$ (Madronich and Calvert, 1990; Donahue and Prinn, 1990; Wallington et al., 1992; Tyndall et al., 2001). However, because ambient $RO_2$

concentrations rarely exceed ambient $HO_2$ (Mihelcic et al., 2003; Tan et al., 2018), reactions with other $RO_2$ will only be





a major $RO_2$ loss pathway in the atmosphere when the rate coefficient equals or exceeds $10^{-11}$ cm$^3$ molec$^{-1}$ s$^{-1}$. This is in contrast to low-NO$_x$ chamber experiments, where $RO_2$ concentrations can greatly exceed $HO_2$ concentrations, potentially biasing the chambers away from reactions important in the atmosphere.

At least for highly oxygenated peroxy radicals, association to form dimers has been found in some experimental studies to proceed with a rate coefficient near the collisional limit ($k > 10^{-10}$ cm$^3$ molec$^{-1}$ s$^{-1}$) (Berndt et al., 2018; Molteni et al., 2019). However Zhao *et al.* find that the typical rate coefficient for presumably similar $RO_2$ + $RO_2$ association reactions is at least an order of magnitude slower (Zhao et al., 2017). These findings may or may not be directly in conflict given the wide range of peroxy-radical rate coefficients. The general tendency is that electron withdrawing groups (i.e. acyl groups in peroxy acyl radicals) increase the $RO_2$ self reaction rate coefficients while electron donating groups (i.e. t-butyl in t-butyl peroxy radicals) lower the rate coefficients. Thus, very fast cross-reaction rate coefficients for highly oxygenated $RO_2$ radicals is consistent with the general trend.

Auto oxidation includes a unimolecular hydrogen shift reaction in which a peroxy moiety abstracts a hydrogen elsewhere on the molecule, generating an alkyl radical, which molecular oxygen will add to in turn, regenerating a peroxy radical functionality on a now more-oxidized carbon backbone. This process is significant in its ability to rapidly generate multi-functional oxygenation products in only a single "generation" of chemistry (stable molecule to stable molecule). However, experimental and theoretical evidence suggests that not every peroxy radical is readily capable of undergoing auto-oxidation. The fraction of peroxy radicals that will auto-oxidize will be different depending on the precursor of interest; for $\alpha$-pinene three of the four peroxy radicals from ozonolysis can auto-oxidize. Ehn *et al.* found that the yield of highly oxidized peroxy radicals from OH oxidation of $\alpha$-pinene was lower than from ozonolysis, indicating that while OH radical-derived peroxy radicals can auto-oxidize, the process is either slower or there are fewer that auto-oxidize readily (Ehn et al., 2014).

To summarize, three important characteristics of a "strongly auto oxidizing" system such as O$_3$ + $\alpha$-pinene are emerging. First, most of the products under atmospheric conditions are likely to be "traditional" oxidation products formed without an auto-oxidation step. Second, the amount of auto oxidation is likely to be strongly temperature dependent, with more at higher temperature. Third, some termination of at least the highly oxygenated $RO_2$ results in covalently bound (and highly oxygenated) dimers, which will have extraordinarily low vapor pressures.

Each consequence has an important implication, which we give in reverse order. First, the dimerization may be the rate-limiting step for "pure organic" nucleation; because it involves a cross reaction, this means that organic nucleation may also be highly sensitive to interactions between different hydrocarbon oxidation sequences, especially at low NO$_x$. It remains somewhat unclear how important this termination is in the real atmosphere vs chambers, given the high $RO_2$:$HO_2$ in chambers; however, the characteristic dimer products are observed in the atmosphere. This strongly suggests that these association reactions are important in the atmosphere.

Second, the temperature dependence of the monomer auto-oxidation products may well be counterbalanced by the simple temperature dependence of vapor pressures. Both of these govern organic condensation, especially to very small particles. Consequently, particle growth rates from organic oxidation (as well as secondary organic aerosol mass yields) are likely to be less sensitive to temperature than one would otherwise expect. They are also likely to be less sensitive to interactions between





different hydrocarbon oxidation sequences. Particle nucleation and growth thus may be partially decoupled, making treatment of each separately important, especially in the "valley of death" region below 3-5 nm where highly diffusive particles are most vulnerable to coagulation loss.

Third, the large majority of organic oxidation products are likely to be oxygenated species with lower vapor pressure than the precursor that remain in the gas phase ("traditional" SVOCs). Later-generation chemistry of these products remains an important topic that can not be forgotten (Donahue et al., 2005, 2012a); these subsequent steps have already been shown to themselves be efficient sources of HOMs (Schobesberger et al., 2013; Ye et al., 2018).

## 3 The model

For this work we will model an idealized batch reactor with oxidation of $\alpha$-pinene leading to HOM products. The model is split into two pieces: a general chemical mechanism and a specific implementation for the batch reactor. Here we only simulate the gas-phase chemistry, culminating in formation (and loss) of $RO_2$ species, with $RO_2$ termination mapping into the 2D-VBS via a succession of kernels specific to each termination step.

### 3.1 Chemical mechanism

The model we use here is an explicit gas-phase photochemical box model using a custom written perl mechanism parser that produces Matlab code for the coupled differential equations describing an arbitrary set of input reactions. The reaction set here is relatively standard small-molecule $HO_x$ chemistry with added $\alpha$-pinene. We list the full set of reactions and rate coefficients in the supplemental material.

The reactions begin with oxidation of a precursor, in this case $\alpha$-pinene via ozonolysis, OH, and the nitrate radical (when $NO_x$ is present). Most of the products of these oxidation reactions will be peroxy radicals, with 10 carbons each, which will undergo two types of reactions: auto-oxidation or radical termination. Auto-oxidation, as described previously, is a unimolecular reaction followed by $O_2$ addition that generates a more oxidized peroxy radical that will have the same reaction pathways available to it again as the parent $RO_2$. It is thus effectively recursive. The radical termination reactions available to a peroxy radicals are reaction with $HO_2$, NO, or another peroxy radical.

It is likely that peroxy radicals have a diversity of H-transfer activation energies (and A-factors), and that this diversity is important to the overall system behavior. We currently lack sufficient kinetic information to constrain this, and so we adopt a simple approach. For the initial $RO_2$ (without additional -OOH functionality), we assume that some fraction can undergo auto oxidation and the remainder cannot (it is a binary choice). All of the $RO_2$ that can auto oxidize does so with a single barrier height and A-factor, and this is common to all of the subsequent oxidized $RO_2$.

A second complication is the role of alkoxy radicals (RO), which can be chain propagating (including a much more widely recognized 1-4 H shift that is broadly similar to the H shift in auto-oxidation) (Atkinson, 1997; Lim and Ziemann, 2009) However, for this set of simulations only we shall assume that RO radicals effectively terminate to molecular products; that is consistent with our broad objective to upgrade the 2D-VBS from simply generating ($NO_x$ dependent) products from a precursor





(i.e. $\alpha$-pinene) to allow the products to be controlled by peroxy radicals. In the future we shall relax this assumption and permit regeneration of $RO_2$ from RO.

### 3.1.1 $\alpha$-pinene reactions

We are simulating $\alpha$-pinene ozonolysis but this chemistry will produce OH radicals and, with added $NO_x$, $NO_3$ as well. Consequently, we have three oxidants to consider. It has been found that OH oxidation of $\alpha$-pinene produces far fewer HOMs than $O_3$ (Kirkby et al., 2016), presumably because the OH addition does not immediately cleave the 6-member ring in $\alpha$-pinene. Further, oxidation of $\alpha$-pinene by $NO_3$ has been shown to produce very little first-generation SOA (Fry et al., 2014; Ng et al., 2017). However, almost all primary oxidation results in peroxy radicals (with the exception of some small fraction of stabilized Criegee Intermediates), though only some of those may have rapid auto-oxidation pathways. To address this, we assign two first-generation $RO_2$ radicals to each oxidation pathway; one, $Ox_0RO_2$, has the potential for auto-oxidation, while the second, $RO_2$, does not.

$$\alpha\text{-pinene} + \text{Ox} \rightarrow \alpha_{ox}\, Ox_0RO_2 + (1 - \alpha_{ox})\, RO_2 + y_{ox}\, \text{OH}$$

Here we assume that $\alpha_{O_3} = 0.25$, $\alpha_{OH} = 0.1$, and $\alpha_{NO_3} = 0$. Further, $y_{O_3} = 0.8$ (Presto and Donahue, 2004), $y_{OH} = 0$, and $y_{NO_3} = 0$.

### 3.1.2 Auto oxidation

The auto-oxidation process involves an internal H-atom transfer in an $RO_2$ radical, immediately followed by $O_2$ addition to reform a new $RO_2$ radical with an added -OOH group. Broadly, we label these reactions

$$Ox_nRO_2 \rightarrow Ox_{n+1}RO_2 \qquad k = A\exp(-\theta_a/T)$$

where $n$ is the number of -OOH groups (the number of generations of auto-oxidation). The rate-limiting kinetic parameter is the unimolecular rate constant for H-atom transfer, which we express as an Arrhenius rate coefficient with a pre-exponential $A = 10^8$ s$^{-1}$ for the $Ox_0RO_2$, which decreases to $A = 7 \times 10^7$ and $A = 6 \times 10^7$ for the $Ox_1RO_2$ and the $Ox_2RO_2$ respectively, and an activation energy $\theta_a = 7500$ K that is consistent for all auto-oxidation steps in our base-case simulation. This gives an H-atom transfer rate coefficient for $Ox_0RO_2$ of $0.01$ s$^{-1}$ at 298 K and $10^{-7}$ s$^{-1}$ at 248 K for the $Ox_0RO_2$. Raising the activation energy to $\theta_a = Ea/R = 8000$ K gives an H-atom transfer rate coefficient of $2 \times 10^{-4}$ s$^{-1}$ at 298 K and $7 \times 10^{-7}$ s$^{-1}$ at 248 K. Even though each H-atom transfer step proceeds with different rate coefficients, we are still likely missing a large amount of the diversity of barriers and rate constants. In addition it is plausible that the barriers drop with increasing extent of auto oxidation.

There are multiple peroxy radical structures formed both initially and via every auto-oxidation reaction. Furthermore, HOM $RO_2$ radicals with multiple -OOH groups can rapidly interconvert, and even the exact same peroxy radical can react with other





radicals to produce different product structures. This leads to a large and complex distribution of final product structures. In order to reduce this complexity, we represent the peroxy radicals as individual surrogate molecules – points in the 2D-VBS space – with auto-oxidation moving surrogate to lower volatility and higher O:C. We thus explicitly represent the volatility of the $RO_2$ radicals, which we show in Figure 1 as a sequence of four colored circles (along with $\alpha$-pinene itself as a red circle),

5 ranging from light green for all "traditional" $RO_2$ that have not undergone auto-oxidation through a succession of deeper green colors for $Ox_nRO_2$ that have undergone $n = 1\text{-}3$ generations of auto-oxidation.

### 3.1.3 $RO_2$ cross reaction kinetics

Based on the available evidence that $RO_2$ self reaction rate constants accelerate with the presence of electron withdrawing functional groups near the -OO moiety, we assume that the self reaction rate constants for the $Ox_nRO_2$ are given by

| $RO_2$ | $k_{n,n}$ |
| --- | --- |
| | $cm^3$ molec $^{-1}$ s$^{-1}$ |
| $RO_2$ | $1.0 \times 10^{-13}$ |
| $Ox_0RO_2$ | $1.0 \times 10^{-13}$ |
| $Ox_1RO_2$ | $1.0 \times 10^{-12}$ |
| $Ox_2RO_2$ | $1.0 \times 10^{-11}$ |
| $Ox_3RO_2$ | $1.0 \times 10^{-10}$ |

Given the enormous number of possible cross reactions (and paucity of kinetic data) Madronich (Madronich and Calvert, 1990) proposed a simple parameterization to estimate the cross-reaction rate coefficient for two dissimilar $RO_2$ radicals as the geometric mean of their self reaction rate constants, $k_{n,m} = 2 \sqrt{k_{n,n} k_{m,m}}$. The factor of 2 is due to the different symmetry of the self and cross reactions.

### 15 3.1.4 Unimolecular $RO_2$ termination

Peroxy radicals may undergo unimolecular termination at any point in the auto-oxidation chain, but exact barrier height and pre-factors for these reactions remain unknown. For example, radical termination may occur via OH loss following a H-abstraction from a carbon with an -OOH group attached. Assuming that the termination involves a bond scission and potentially an H-transfer between neighboring groups, this termination is likely to have a higher A-factor but a higher activation energy that

20 the internal isomerization (which has a long cyclic transition state and thus a relatively low A-factor). Here we represent this process with a pre-factor of $10^{15}$ s$^{-1}$ and a barrier of 13000 K. This gives a termination rate coefficient of $10^{-4}$ s$^{-1}$ at 298 K and $1.7 \times 10^{-8}$ s$^{-1}$ at 248 K. As with auto-oxidation, there is every reason to believe that different functional groups on the peroxy radical would hinder or enhance the unimolecular termination pathway, but we use a consistent pre-factor and barrier for every peroxy radical to represent a complicated reaction pathway more simply.





25     In our framework, unimolecular termination products are treated as "monomer" products previously discussed. They may or may not be HOMs depending on whether the peroxy radical that produced them underwent auto-oxidation to any extent.

### 3.1.5    $RO_2$ dimerization

The mechanism for $RO_2$ dimerization is unknown. It is a theoretical puzzle because $RO_2$ reactions are thought to proceed through a weakly bound "tetroxide" intermediate, ROO-OOR, which is a singlet closed-shell species (Lee et al., 2016). If by

whatever (presumably multiple step) process this singlet species eliminates molecular oxygen, $O_2(^3\Sigma)$, the singlet peroxide ROOR is spin forbidden. In the more common radical pathway, which produces $RO + RO + O_2$, the two RO radicals are spin entangled in a triplet state. They cannot re-combine into a peroxide (ROOR) without first crossing to the singlet state.

    Here we model the $RO_2$ cross reactions as

$$RO_2 + R'O_2 \rightarrow \gamma\, ROOR' + (1-\gamma)\,(RO + R'O) + O_2$$

with "products" being treated as stable molecules within the Volatility Basis Set. A more general treatment would also include both the radical (RO) and molecular (alcohol + carbonyl) pathways, which are treated with a branching ratio $\beta$ (Tyndall et al., 2001); here we assume that $\beta = 1$ for all of these peroxy radicals, consistent with findings that the radical pathway is favored for peroxy radicals with electron withdrawing functional groups near the -OO moiety.

    It may be that the ability to form dimers is directly dependent on additional functional groups in the $RO_2$ molecules, or it

may be an indirect consequence, for example related to volatility and possibly cluster lifetime when two $RO_2$ radicals interact. Because we are explicitly tracking the volatility of the $RO_2$ radicals produced via auto oxidation (and because some have very low volatility), here we shall explore the possibility that dimerization is somehow tied to the volatility of the reacting $RO_2$ radicals. Specifically, we hypothesize that when low volatility $RO_2$ species collide they will form a short-lived (radical) cluster, both before and after reacting. We further assume that ROOR dimer formation in this cluster follows the more traditional

radical (RO) step, and that the product cluster must be long enough lived for collisions with air molecules to induce a spin flip in the initially triplet entangled system. As a rough measure of cluster lifetime we will use the geometric mean volatility of two interacting $RO_2$ species (i.e. the average of the $\log C^\circ$ values).

    If dimerization is volatility dependent, the probability of dimerization occurring when two peroxy radicals interact may be approximated by:

$$\gamma = \frac{1}{1 + C^\circ_{\text{GM}}/C^\circ_{\text{ref}}} \qquad\qquad (1)$$

where $C^\circ_{\text{GM}}$ is the geometric mean of the effective saturation concentrations of the two reacting peroxy radicals and $C^\circ_{\text{ref}}$ is a temperature dependent reference saturation concentration representing where we expect half of the reacting peroxy radicals to dimerize. In our base case we assume a rate coefficient for highly oxidized $\alpha$-pinene peroxy radical reactions near the





collisional limit, but given the branching ratio in Eq. 1 the dimerization product yield can be much lower (depending on $RO_2$ volatility).

## 3.2 $RO_2$ termination kernels and the VBS

Following $\alpha$-pinene oxidation, the initial, "traditional" $RO_2$ is a di-carbonyl peroxy radical ($C_{10}H_{15}O_4$). Each carbonyl functional group reduces the volatility by about one order of magnitude (Pankow and Asher, 2007), and we assume that the -OO moiety decreases the volatility by roughly another 1.5 orders of magnitude; consequently, this $RO_2$ has a $C^\circ(300) \simeq 10^4$ $\mu$g m$^{-3}$. Each successive $Ox_nRO_2$ differs from its predecessor by having an additional -OOH functional group. These nominally decrease volatility by 2.5 orders of magnitude (Pankow and Asher, 2007); however, there is evidence that multifunctional molecules with opportunities for internal hydrogen bonding have higher volatilities than simple composition activity relations suggest (Kurtén et al., 2016). To crudely represent this we assign $C^\circ(300) \simeq 10$ $\mu$g m$^{-3}$ for $Ox_1RO_2$ but assume that each successive generation of auto-oxidation decreases volatility by only 2 orders of magnitude, as shown in Figure 1. It is interesting to note that $Ox_2RO_2$ on up the auto-oxidation sequence have a low enough volatility that they are likely to remain in particles if they collide with them.

Figure 1 also shows the broad "XVOC" ranges as colored bands. This includes the new ULVOCs in purple, which have a sufficiently low volatility ($C^\circ(T) \leq 3 \times 10^{-9}$ $\mu$g m$^{-3}$) to nucleate efficiently under typical conditions (it is supersaturation that ultimately drives this). The ELVOCs, in contrast, will stick to any particle of any size they hit, but may not contribute significantly to nucleation itself (in practice it may be the geometric mean of $C^\circ(T)$ for two colliding vapors that governs nucleation, just as we hypothesize for ROOR formation).

We represent $RO_2$ termination reactions to products as kernels anchored to the peroxy radical point that produced them. These kernels allow us to represent a wide variety of stabilization pathways producing a wide variety of different products through one surrogate species; the variety of species that the surrogate represents are instead mapped to a distribution of products within the 2D-VBS defined by a transformation relative to the surrogate $RO_2$ volatility and O:C. In Figure 1 we show the sum of these kernels (weighted by branching) for the three broad classes of products defined earlier: "traditional" oxidation products, monomers involving at least one generation of auto oxidation, and ROOR dimers. We provide the individual kernels (relative to the $C^\circ$ and O:C values for each $RO_2$) in the supplemental material. The net effect is a concentration of traditional products with O:C modes in the SVOC and IVOC range, which in earlier VBS parameterizations of $\alpha$-pinene SOA have constituted all of the products (Presto and Donahue, 2006; Jimenez et al., 2009; Donahue et al., 2012b), augmented by a tongue of HOM monomer products extending through the LVOC range and up to O:C > 1, and finally with a parallel shoal of HOM dimer products spanning the ELVOC range and extending into the ULVOC range (causing "pure biogenic" nucleation (Kirkby et al., 2016)).

We fully resolve the reaction products and then take the final concentration of every surrogate species (for example "$Ox_2RONO_2$" represents all organonitrates derived from second-generation auto-oxidized peroxy radicals) and map those to a distribution of products in the 2D-VBS using the appropriate kernel. This can readily be adapted to form a module (operator) within a larger framework representing particle microphysics, transport, wall loss, etc.





### 3.3 Ideal reactor

Here we model an ideal batch reactor with an initial input of precursor $\alpha$-pinene that is oxidized over time leaving only stable oxidation products, with essentially no precursor remaining at the end of the simulation. There is no wall loss, ventilation, additional vapors added after the initialization, nor particle condensation or nucleation; our goal here is to just probe the chemistry of interest without interference of the physics of typical chamber experiments. We present results varying the temperature, $NO_x$ and barrier-height dependence of the yields of $\alpha$-pinene ozonolysis. We simulate 600 ppt of $\alpha$-pinene ($1.5 \times 10^{10}$ molec cm$^{-3}$) reacting with 40 ppb of ozone, which corresponds to a lifetime of approximately 4 h ($1.4 \times 10^4$ s) at 298 K. When we simulate $NO_x$ chemistry we add the $NO_x$ to the system as NO, after which it undergoes reaction to $NO_2$ and reaches an equilibrium ratio of $NO:NO_2 \simeq 1:10$. The value we report as the concentration of $NO_x$ is the amount of NO initially added into the system.

## 4 Results

We present results showing the final products of the chemistry described above and how they resolve within the 2D-VBS. We make qualitative comparisons with previous experimental results to show the validity of our approaches.

### 4.1 Base-case run

We will start with a "base-case" run at 298 K without $NO_x$ to demonstrate the model and its output. As the $\alpha$-pinene reacts, the peroxy radicals quickly build up to a steady state and then decay in consort with the $\alpha$-pinene as they auto-oxidize and form stable products. As Figure 2 shows, each of the successively more oxidized peroxy radicals reaches a maximum concentration at a similar time, but to progressively lower maximum values. The stable products are formed via the various peroxy radical termination processes and in this simulation they have no sinks and so simply accumulate. Their yields are thus simply their final concentrations divided by the initial $\alpha$-pinene concentration.

In Figure 3a we show one such set of terminal products, labeled $Ox_n$ to indicate the number of auto-oxidation steps. These products form from both $RO_2 + NO$ and $RO_2 + RO_2$ reactions and so scale with the $RO_2$ radicals themselves, but because they are not lost in the batch reaction (here we do not simulate aging with the 2D-VBS) their concentrations accumulate over time.

In Figure 3b we show the corresponding dimer (ROOR) concentrations. We identify the dimers by a number pair $(n, m)$ indicating the auto-oxidation extent of the associated $Ox_n RO_2$. Here the situation is more nuanced because the dimers are formed exclusively through cross reactions and we assume (Eq. 1) that the dimer yields depend on the $RO_2$ volatility. Consequently, even though the most oxidized $RO_2$ have lower concentrations, they produce dimers with higher yields. The 2-2 and 3-3 symmetric dimers, along with the 2-3 cross dimer, are the most abundant.

The ultimate product distribution is dependent on the competition between auto-oxidation and termination via other radical species. Thus it is important to fully implement $HO_x$ and $NO_x$ chemistry and it is informative to see how the concentrations of these radicals vary throughout the simulation. We show $HO_2$ and NO for various levels of added $NO_x$ in Figure 4, along with





the sum of oxidized RO$_2$ ($\sum$ Ox$_n$RO$_2$). Characteristic for chamber experiments, HO$_2$ remains at a low level ($\lesssim 10^7$ molec cm$^{-3}$) compared to the atmosphere due to the relative scarcity of species such as CO and CH$_2$O that directly convert OH into HO$_2$ and also photolyze to produce HO$_2$. Thus HO$_2$ remains far rarer than even the OxRO$_2$ while most $\alpha$-pinene is oxidized, with OxRO$_2$:HO$_2$ $\simeq$ 40 and all RO$_2$:HO$_2$ $\simeq$ 80. Because in our simulation we add NO$_x$ as NO, at first the NO levels drop rapidly via reaction with HO$_2$ and RO$_2$, but then reach a steady state as NO$_2$ photolysis replenishes NO. In general for batch chamber experiments, NO$_x$ is not added at the steady-state NO:NO$_2$, and it is common to add NO as we simulate here; the rapidly evolving NO concentrations complicate interpretation of the results as the conditions can move from "high NO" early in the experiment to "low NO" later on. However, in this simulation, for 100 pptv added NO$_x$ the NO stabilizes near 3 pptv. This is sufficient to slightly suppress the HO$_2$, more significantly reduce the oxidized RO$_2$, and to dominate RO$_2$ termination.

The OxRO$_2$ curves in Figure 4 allow one to assess RO$_2$ termination in general, remembering that the rate coefficients for HO$_2$ and NO termination are similar and (in our model) the OxRO$_2$ self reaction rate coefficients are up to ten times faster. This means that HO$_2$ is never an important RO$_2$ sink under these conditions, while NO becomes competitive between 50 and 100 ppt added NO$_x$ (the NO concentration must be 10 times higher than the OxRO$_2$ to compete). Increasing NO significantly suppresses OxRO$_2$ and so by 1000 ppt NO$_x$ the NO pathway is dominant at all times.

## 4.2 Temperature dependence

The temperature dependence of the HOM formation chemistry is a key diagnostic with great atmospheric significance. Experimental data indicate that HOM yields decrease with decreasing temperature and that HOM dimers especially follow this trend. This does not mean that condensible LVOC products necessarily decrease as products with a higher $C^\circ(300)$ will condense when it is cold; the XVOC color patches in the 2D-VBS figures show this volatility temperature dependence. However since essentially all of the (room temperature) ULVOCs and ELVOCs are dimers, their yields will follow the dimer trends.

In Figure 5 we show the temperature dependent results for our base-case NO$_x$-free simulation. Each column is a different temperature (248, 278, and 298 K). The top row is a 2D-VBS contour plot of the products, recapitulating Figure 1 with the volatility classes (defined by $C^\circ(T)$) shifting with temperature following Stolzenburg (Stolzenburg et al., 2018). The middle row is a 1D-VBS plot (summing the top plots over O:C) with histograms of carbon yields in each volatility bin, colored by the RO$_2$ termination process (NO, dimerization, HO$_2$, etc). The bottom row is color coded by the generation of auto oxidation, with darker colors corresponding to products of RO$_2$ that have undergone more auto-oxidation steps.

In the bottom row of Figure 5 we can clearly see that the amount of products formed from auto-oxidized peroxy radicals decreases with decreasing temperature to the point where at 248 K, there is no significant contribution of auto-oxidation to the yields at all. This is expected from the temperature dependence of the auto-oxidation rate coefficient and the trend is supported with experimental data. In the middle plots, we see a reduction in HOM monomers at lower temperatures, but because the volatility classes shift toward higher $C^\circ(300)$ at lower temperature, there are still relatively high LVOC yields, which is consistent with previous growth rate measurements indicating large contributions from LVOCs to growth even at low temperature. We also see that at lower temperatures dimer yields are greatly reduced. However, at 248 K, we see an emergence of non-HOM dimers, dimers formed via reaction between two non-auto-oxidized peroxy radicals. These are "non-





HOM" dimers as the definition of "HOMs" requires there to be an auto-oxidation step in the formation process. They are a consequence of our hypothesis that dimer formation is related to $RO_2$ complex lifetime, as expressed in Eq. 1; any temperature dependence of dimer yields (and their O:C) is thus a test of this hypothesis.

In addition, we see almost no formation of products from an $RO_2$ + $HO_2$ pathway. This is not a major pathway for peroxy radical termination at any temperature, given our base-case assumption of a fast $RO_2$ + $RO_2$ reaction and the high $RO_2$:$HO_2$ in low-$NO_x$ chamber conditions. There is however a slight temperature dependence of hydroperoxide formation as at low temperatures (with no $NO_x$) there is a larger build-up of the $OX_0RO_2$, which have a relatively slow self reaction rate constant and thus do terminate with $HO_2$.

**4.3  $NO_x$ dependence**

Experimental data indicate that ULVOC/ELVOC yields are strongly reduced with increasing $NO_x$ while LVOC and SVOC yields are less affected. In addition, at intermediate $NO_x$ concentrations, HOM organo-nitrates will be formed contributing HOMs in the SVOC to LVOC range not found under low-$NO_x$ conditions.

    We show the $NO_x$ dependent results for our 298 K simulation in Figure 6, which follows the temperature-dependent Figure 5
in form. Each column is a different $NO_x$ concentration (50, 100, and 1000 ppt), but the results show the same qualitative pattern as those in Figure 5, with high $NO_x$ taking the place of low temperature. NO suppresses auto oxidation and HOM formation just as low temperature does, but "normal" hydroperoxides (formed via $HO_2$ termination) are replaced by organonitrates. Further, because the temperature is not reduced, the high-$NO_x$ products remain largely in the SVOC class, so high $NO_x$ will suppress nucleation.

At very low $NO_x$ ($< 20$ ppt), we see relatively little effect on yields, either type or auto-oxidation, compared to the zero-$NO_x$ 298 K results in Figure 5. There is, however, some suppression of dimers. In our simulation, $NO_x$ becomes competitive with auto-oxidation around 500 ppt and completely overpowers auto-oxidation around 1000 ppt. This is based on an $RO_2$ + NO rate coefficient of $10^{-11}$ cm$^3$ molec$^{-1}$ s$^{-1}$ and an activation barrier to auto-oxidation producing at rate coefficient of about $10^{-3}$ s$^{-1}$ at 298 K. In this simulation we do not incorporate any variation in kinetics with the $RO_2$ classes (e.g. the isomerization
activation energy) other than the dependence of dimerization on the geometric mean of $RO_2$ volatility expressed in Eq. 1 and the small change in the pre-factor for auto-oxidation discussed in section 3.1.2. Therefore we caution against making quantitative comparisons to experimental results such as Zhao et al., who found that competition between auto-oxidation and $NO_x$ termination begins to favor the $NO_x$ pathway at about 20 ppbv of $NO_x$. It is highly likely that the $RO_2$ kinetics vary considerably depending on the exact $RO_2$ structure and very possible that more of this will need to be represented to obtain
quantitative agreement with experimental observations.

    In Figure 7 we show the $NO_x$ effects on HOM monomers and dimers at different temperatures. For the $NO_x$ range we simulate here, the major effect of NO is to suppress the total HOMs uniformly by roughly a factor of 10. In this plot we see that while temperature has a relatively uniform effect of suppressing HOM monomers and dimers, $NO_x$ suppress dimer yields more aggressively than it does HOM monomer yields. This can be explained due to the emergence of HOM nitrates, which are





included in the monomer yields, as well as the consequence of NO reducing total OxRO$_2$ and thus the rate of peroxy radical association reactions.

It is important to note that the relatively low HO$_2$ concentrations in these runs may be enhancing the importance of NO$_x$ chemistry here. With higher HO$_2$ concentrations, the RO$_2$ + HO$_2$ reaction will compete with the RO$_2$ + NO and reduce the nitrate yields. However, the HO$_2$ concentrations simulate here line up with what is often seen in very clean chamber experiments and thus our NO$_x$ conclusions hold for those cases.

### 4.4 Sensitivity studies

As stated, the energy barrier to auto-oxidation is an important unknown in this model. It is constrained experimentally by the competition with the RO$_2$ + NO reaction, but the coefficient associated with auto-oxidation could span orders of magnitude for different peroxy radicals. In all the simulations described so far, we used a single barrier for auto-oxidation for every peroxy radical, and reduced the pre-factor as $n$, the extent of auto-oxidation, increased.

We employ two methods to investigate the barrier height dependence look at the yields: fixing the rate coefficient at 298 K and varying its activation energy (in a Clausius-Clapeyron like expression) and allowing the rate coefficient to vary within an order of magnitude of what was used in the "base-case" scenario over the whole temperature range. This is to see the effect of the auto-oxidation barrier height on the HOM and dimer yields as well as to provide a context for the error associated with our assumption regarding the barrier height. The effect of these methods on the auto-oxidation rate constant is shown in the Supplemental Information.

As shown in Figure 8, changing the barrier height strongly affects the calculated amount of auto-oxidation, especially at low temperatures. There is little effect near 298 K because that is where we prescribe the rate coefficient. In Figure 9, we show results from allowing the rate coefficient to vary an order of magnitude over the entire temperature range, producing dimer and monomer yields that span orders of magnitude. The solid curves indicate the base-case scenario described above. This allows us to put some bounds on the rate coefficient of auto-oxidation as experimentally dimer and monomer yields fall well within the shaded region of the plot.

### 4.4.1 NO$_x$ dependence of temperature dependence

We also want to investigate how the similar trends of decreasing temperature and increasing NO$_x$ work together. The bottom plot of Figure 10 clearly shows this synergistic effect on auto-oxidation, as products that have undergone any auto-oxidation are almost completely suppressed at the highest NO$_x$ concentration and the lowest temperature investigated here. The top plot of Figure 10 shows once again that decreasing temperature and increasing NO$_x$ work together to suppress both HOM monomers and HOM dimers as well as a few other trends. As NO$_x$ is increased, the yield of nitrates increases, but at any specific NO$_x$ concentration the amount of nitrate formed is effectively temperature-independent. However, HOM nitrates are only formed at the highest temperature and at lower NO$_x$ concentrations; obviously, to form nitrates some NO$_x$ needs to be present, but if too much is present, the peroxy radicals will terminate before they can auto-oxidize to form HOM nitrates. At low NO$_x$ concentrations the HOM formation is dependent on temperature as discussed previously, but at high NO$_x$ concentrations the

yields are completely dominated by termination with NO. We can also see at low $NO_x$ and low temperatures, the appearance of non-HOM dimers; formation of these less oxidized peroxides is a consequence of our assumption that a long-lived $RO_2$ molecular cluster is required to allow the products to cross to the singlet spin surface. This plot also shows the temperature
and $NO_x$ dependence of yields from the $RO_2 + HO_2$ reaction. While never a large portion of the total yield, products of this reaction increase under low temperature and low $NO_x$ conditions. When the temperature is low, most of the peroxy radicals present are $Ox_0RO_2$, which based on our association rate constants, will react the slowest with each other. This allows for more of the peroxy radicals to react with $HO_2$, even at the relatively low $HO_2$ concentrations present. At high $NO_x$, the $RO_2 + NO$ reactions dominates everything including the $RO_2 + HO_2$ reaction and very few to no ROOH products are formed.

## 5 Conclusions

Here we present a model that represents peroxy radical chemistry semi-explicitly and maps the products of that chemistry onto the 2D-VBS. We investigate the dependence of the product yields on temperature, $NO_x$, and the energy barrier to auto-oxidation. Ultimately both HOM and dimers are suppressed under conditions when auto-oxidation is suppressed via the competition with radical termination processes. Competition with bimolecular termination processes is enhanced when the $RO_2$
lifetime with respect to bimolecular radical termination is shortened (higher radical concentrations) or when auto-oxidation is slower (higher energy barrier). Therefore HOM and dimer production is highest under high temperature, low $NO_x$ conditions and if we assume a lower energy barrier to auto-oxidation. These simulations were conducted assuming relatively fast dimerization rate constants. Therefore should some or all peroxy radicals dimerize slower, another important competition, that is not discussed at length here due to it's relatively low impact, the $RO_2 + HO_2$ reaction, could be much more important.
Overall, our simulation results are consistent with emerging experimental observations and strongly suggest that HOM formation will be strongly temperature dependent under atmospheric conditions, as well as highly sensitive to NO. The new "radical" VBS allows us to explicitly simulate the $RO_2$ termination without making ad-hoc assumptions about "high" or "low" NO conditions and also enables general consideration of $RO_2$ termination chemistry, including cross reactions among $RO_2$ derived from many hydrocarbon precursors. It is thus well suited to describe these rich chemical systems.

*Author contributions.* MS conducted the model runs and wrote the paper; NMD designed the study, advised research progress, and commented extensively on the paper drafts.

*Competing interests.* The authors declare that they have no conflict of interest.

*Acknowledgements.* This work was supported by grant AGS 1801897 from the U. S. National Science Foundation.



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



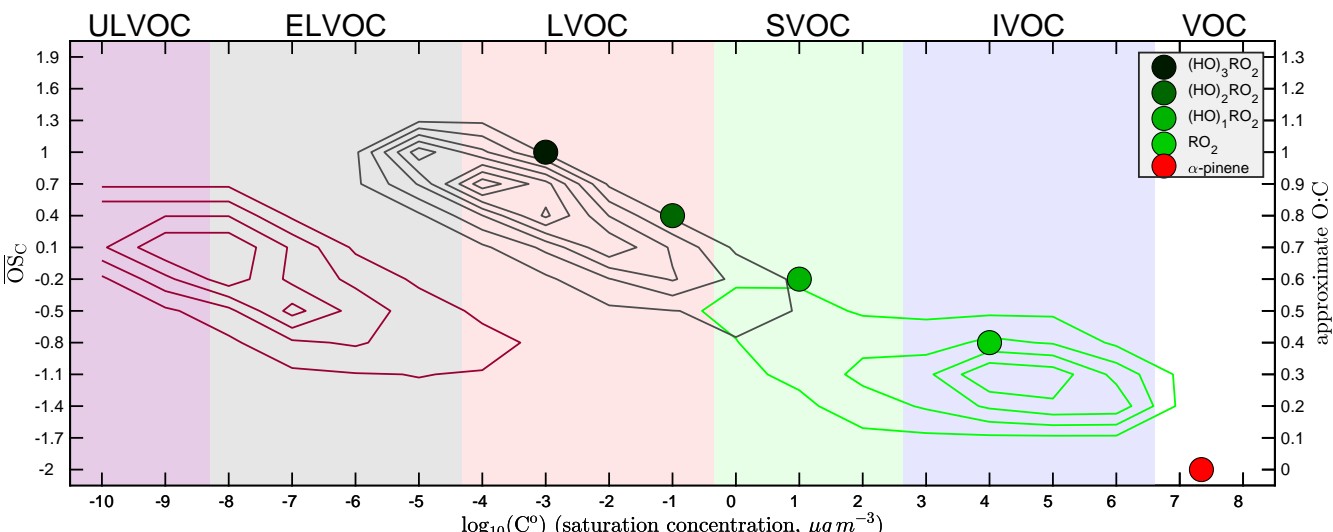

**Figure 1.** Basic $\alpha$-pinene oxidation and the two-dimensional volatility basis set (2D-VBS), including explicitly treated peroxy radicals (RO$_2$). Axes are volatility $\log_{10} C^\circ(300)$) and O:C. Broad colored bands are volatility classes, indicated along the top: ULVOCs are efficient nucleators. $\alpha$-Pinene (red, lower right) oxidizes to produce a succession of peroxy radicals, first a "traditional" radical (RO$_2$, in light green) and then a succession of oxidized RO$_2$ via auto-oxidation (Ox$_n$RO$_2$, darker shades of green). RO$_2$ radicals ultimately terminate into molecular products, represented here by contours depending on whether they derive from the traditional RO$_2$ (green contours), auto-oxidation (C$_{10}$ monomers, gray contours), or Ox$_n$RO$_2$ dimerization (C$_{20}$ ROOR, maroon contours).

**Figure 2.** Concentration time series of peroxy radicals (RO₂). The peroxy radical concentrations over the course of a simulation peak early as they are produced via $\alpha$-pinene oxidation and decrease as they react away over the course of the simulation. This simulation is run at 298 K and with no $NO_x$ although the trends are similar across the temperatures and $NO_x$ concentrations investigated here, with the difference being in the absolute concentrations, especially those of the more oxidized peroxy radicals.



**Figure 3.** Concentration time series of monomer and dimer products. $Ox_n$ indicates the number of auto-oxidation steps. All of these products are formed via peroxy radical chemistry, but the products are still predominantly non-HOM. This trend persists for hydroperoxides and organonitrates when $NO_x$ is present. The dimers do not follow the same trend of forming more from the less oxidized peroxy radical as the other species do. The dimer plot is colored by the sum of how oxidized the two peroxy radicals that reacted to form the dimer are (i.e. the dimerization of an $Ox_1RO_2$ and an $Ox_2RO_2$ would be consider sum of $Ox = 3$). However, they also do not follow the exact opposite trend. This is due to the branching ratio to dimers being larger for low volatility peroxy radicals, but those peroxy radicals being less common leading to mid-range oxidized peroxy radicals forming the most dimers.

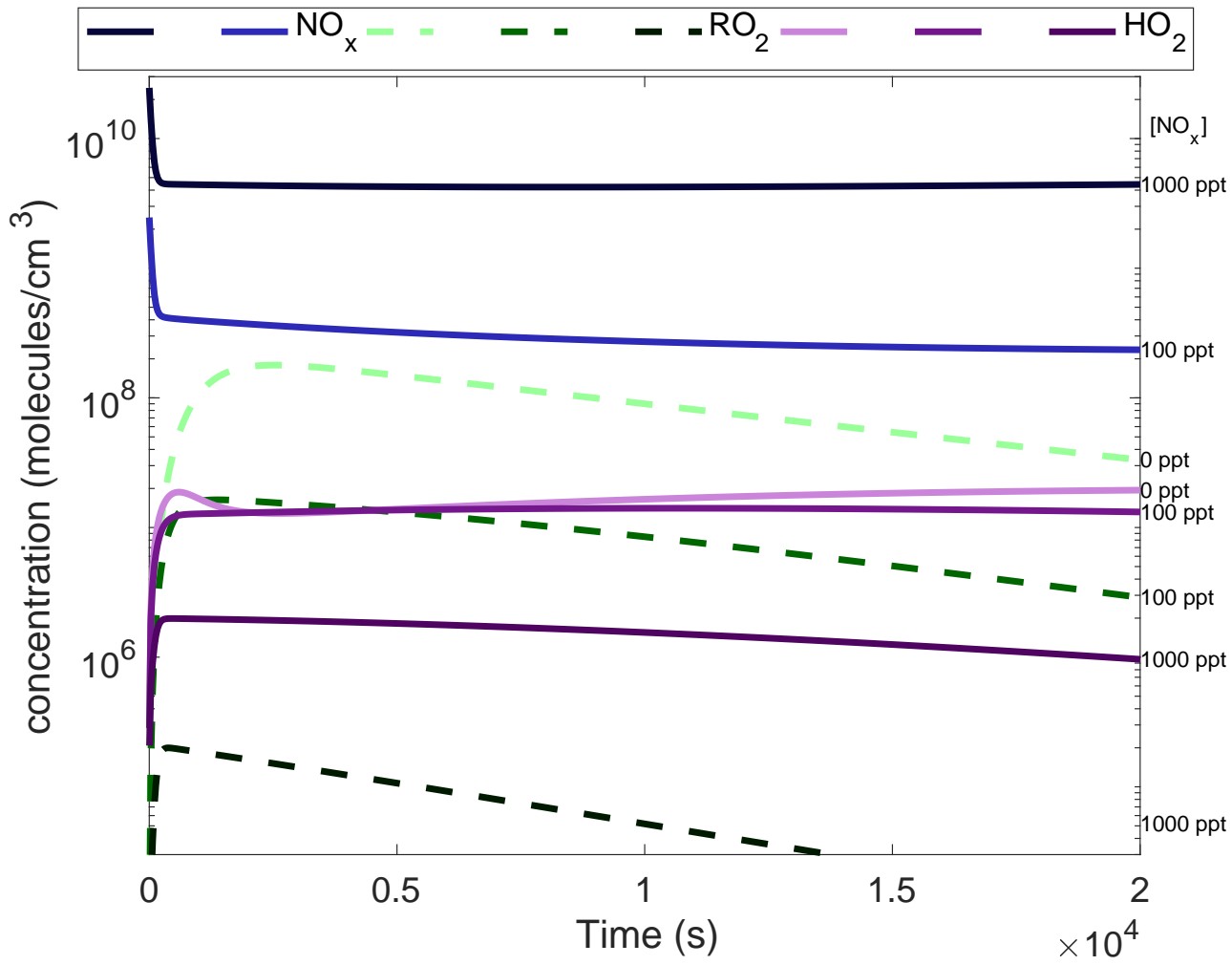

**Figure 4.** Concentration time series of $HO_2$, $NO$, and oxidized $RO_2$ at various $NO_x$ levels. $HO_2$ remains reasonably stable near $10^7$ molec cm$^{-3}$ (0.3 pptv). $NO_x$ is introduced as $NO$ and drops to a steady state as it reacts with $HO_2$ and $RO_2$ to form $NO_2$, which in turn photolyzes. By 1000 pptv added $NO_x$ the $NO$ greatly exceeds the $HO_2$ and somewhat suppresses $HO_2$ concentrations. The oxidized $RO_2$ ($\sum Ox_n RO_2$) reaches a peak just over $10^8$ molec cm$^{-3}$ at zero $NO_x$ before gradually decaying.



**Figure 5.** Yields of $\alpha$-pinene oxidation products without any NO$_x$ present at 248 K, 278K, and 298K. As temperature decreases, in the top, contour plots the dimer products shift to higher volatilities and lower O:C, even to the point where the dimers have approximately the same C*(300K) as the monomer products, which is supported by experimental data. In addition, the diversity of both monomer, and dimer products decreases with decreasing temperature. In the middle row of plots, where we color the yields by what type of product they are, we again see that shift of the products to higher volatilities. In addition, we see fewer dimers formed at lower temperatures. This makes sense based on the hypothesis that when two peroxy radicals react the probability that they formed a dimer is dependent on their volatility, and thus dependent on the extent of auto-oxidation. The dimers formed at the lowest temperature are mostly non-HOM dimers formed from reactions of the initial peroxy radicals, where no auto-oxidation occurred. It is also of note that the formation of hydroperoxides is not a significant fraction of the yield at any temperature due to the low concentration of HO$_2$. The bottom plots, colored by the extent of auto-oxidation the products underwent before terminating, show that as we decrease temperature, the extent of auto-oxidation that the peroxy radicals undergo decreases. At 248 K, we see almost no products that have undergone any auto-oxidation.





| (a) 50 ppt | (b) 100 ppt | (c) 1000 ppt |

**Figure 6.** Yields of $\alpha$-pinene oxidation products without at 298 K with 50, 100, and 1000 ppt of NO introduced into the system. There are similar trends seen in the contour plot with an increase in the volatility of both HOM monomer and dimer products as well as a decrease in the diversity of both with increasing $NO_x$. In the second row of plots, we see an increase in the amount of nitrates formed and the depletion of dimers formed with increasing $NO_x$. Similarly to the temperature trend, as we increase $NO_x$ the products shift to higher volatilities. However, once we reach a high enough $NO_x$ concentration to produce essentially no dimers, we still see some LVOC products that include nitrates. This is consistent with experimental evidence that $NO_x$ suppresses nucleation, but not necessarily growth. From the last plots, we can clearly see fewer auto-oxidation products at higher $NO_x$ concentrations with a nearly complete suppression of auto-oxidation between 500-1000 ppt of $NO_x$.

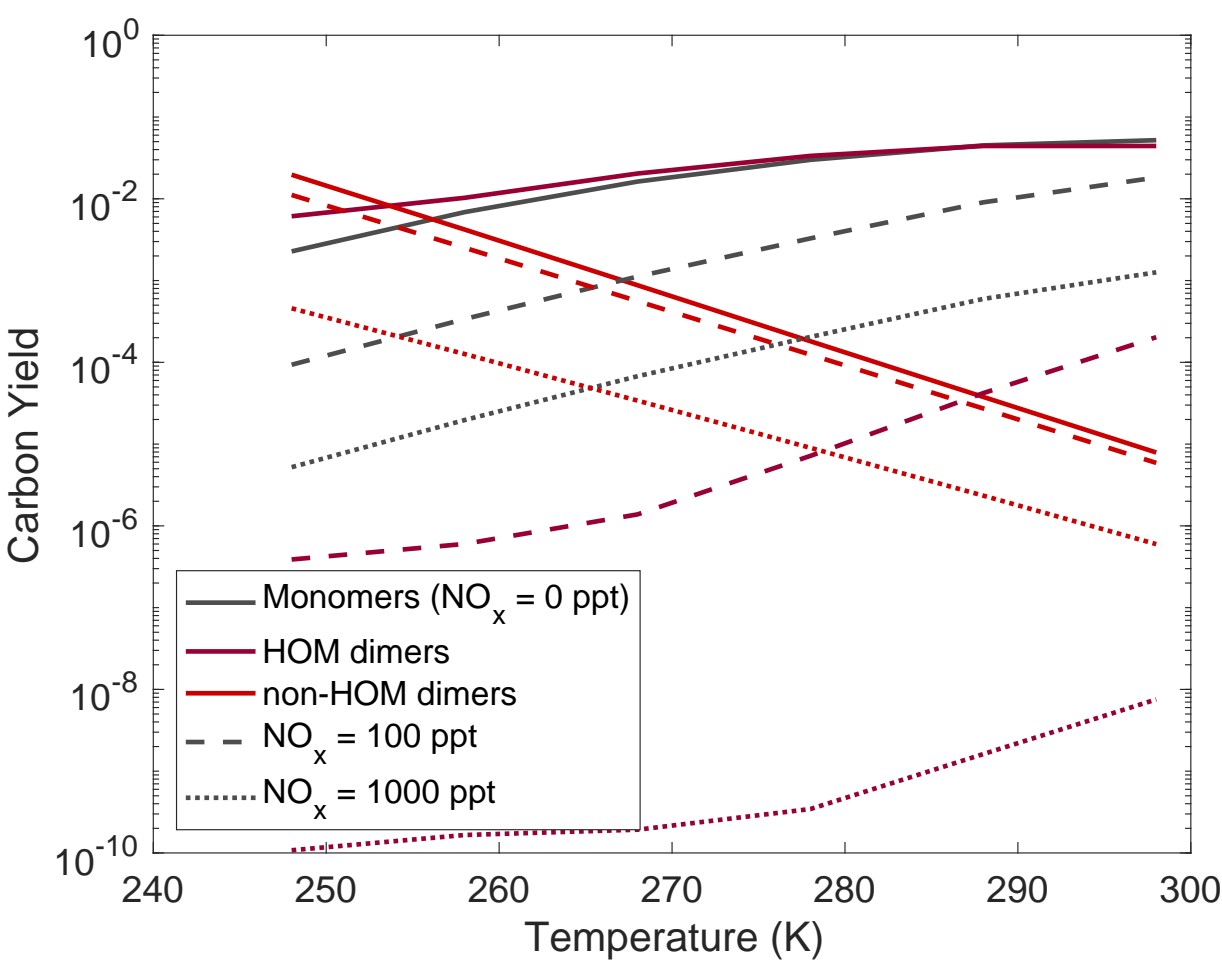

**Figure 7.** Yields at different temperatures and various $NO_x$ levels: no $NO_x$ present (solid curves), 100 ppt of $NO_x$ present (dashed curves) and 1000 ppt of $NO_x$ present (dotted curves). Very low $NO_x$ concentrations have little effect on HOM yields, however $NO_x$ eventually becomes competitive with HOM-producing pathways and there is a significant reduction in both dimer and HOM monomer yields accompanied by an increase in non-HOM dimers. HOM dimers are more strongly affected by suppression of auto-oxidation than HOM monomers.

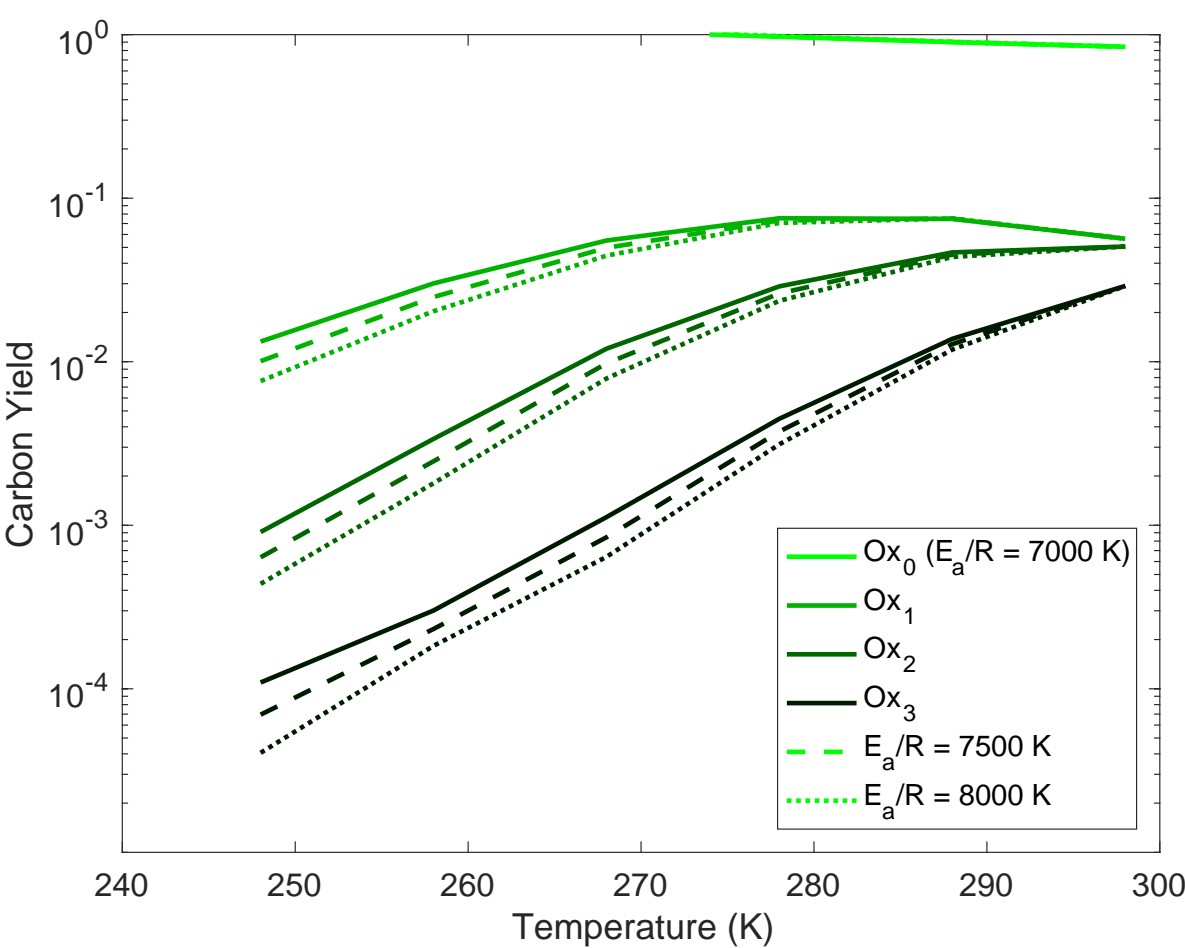

**Figure 8.** Variation in carbon yields with temperature for different auto-oxidation barrier heights. Here the auto-oxidation rate coefficient at 298 K is fixed at $0.01$ s$^{-1}$ and we vary the activation energy. The dashed curve shows the results for the rate coefficient in our base-case simulation. The solid curve has a lower activation energy and thus a weaker temperature dependence, while the dotted curve has a higher activation anergy and thus a stronger temperature dependence. As expected auto-oxidation is suppressed more quickly as temperature drops when the activation energy is higher; this translates to lower HOM yields.

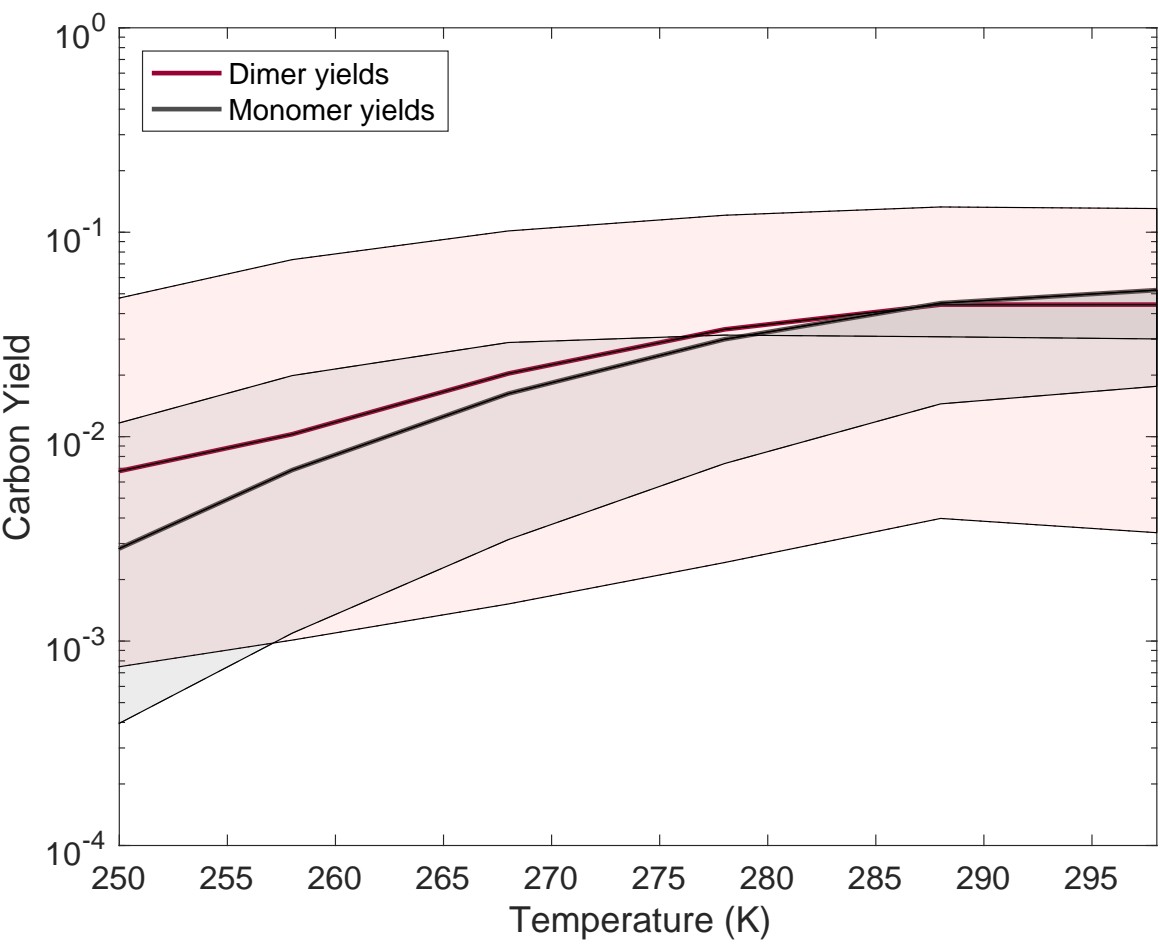

**Figure 9.** Changes to HOM monomer and dimer yields with an order of magnitude variation in the rate coefficient at 298 K by changing the auto-oxidation A factor. The rate coefficient at temperatures other than 298 K is calculated in the same manner as described above. Almost all of the yields at a particular temperature are increased relative to the traditional case when we assume the rate coefficient at 298 K is higher than the traditional case and the yields decrease at a particular temperature when we assume the rate coefficient is lower than the traditional case. The monomer yields decrease with a faster auto-oxidation rate constant at high temperatures because with more auto-oxidation, more association reactions successfully form dimers, thus reducing the monomer yields. Quantitatively, as we vary the rate coefficient of auto-oxidation by an order of magnitude, we are seeing a corresponding increase or decrease in the HOM product also by about an order of magnitude.



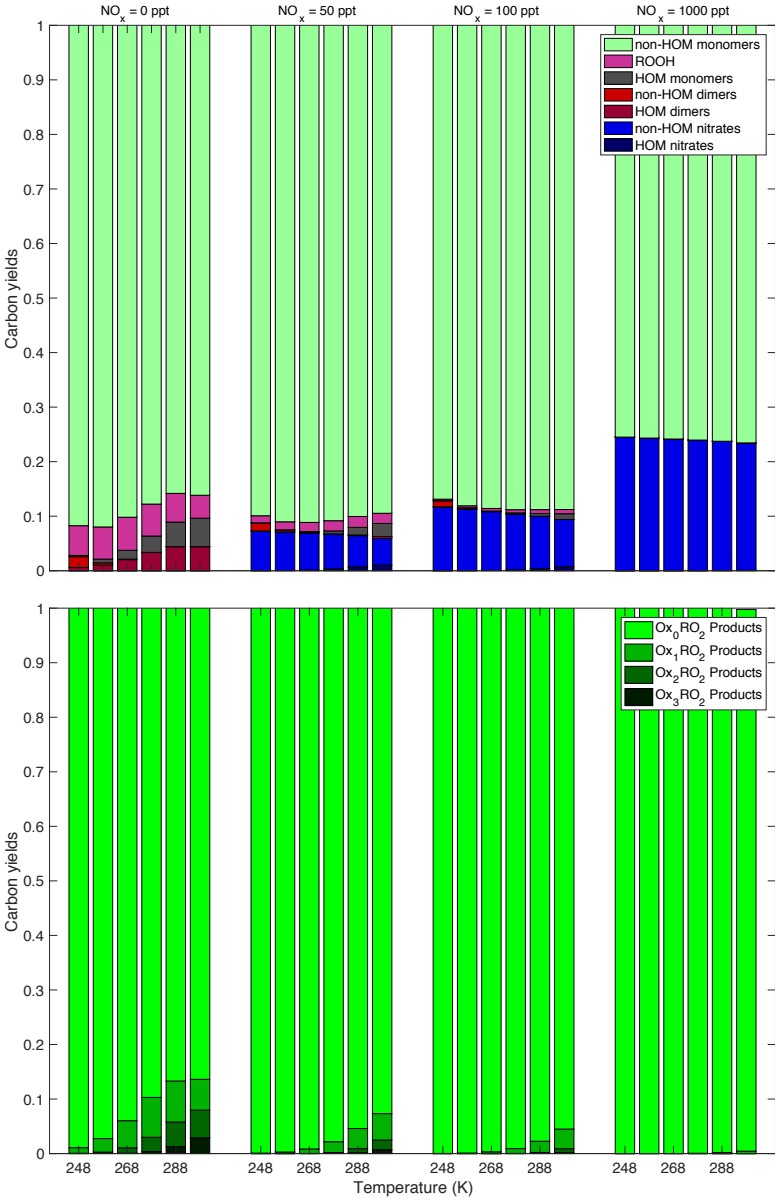

**Figure 10.** This plot shows the synergistic effect of the $NO_x$ and temperature on the product yields. The same trend persists at every $NO_x$ concentration specifically the decrease in the amount of HOMs-both monomers and dimers with decreasing temperature. Of note is that at the highest $NO_x$ concentration shown here (1000 ppt NO added), the HOM production is close to 0 at every temperature. Thus complete suppression of HOMs occurs around this $NO_x$ concentration. This plot shows both the temperature dependence and the $NO_x$ dependence again, but colors are based on how oxidized the peroxy radical that produced the products were. The products represented by the darker colors underwent more auto-oxidation before terminating than those represented by the lighter colors. We can see that auto-oxidation is suppressed both by temperature and by $NO_x$. Once again, we see that at a high enough $NO_x$ concentration, auto-oxidation is effectively suppressed at all the temperatures investigated here.