# Peer review of "Peroxy Radical Chemistry and the Volatility Basis Set"

_Atmospheric Chemistry and Physics, 2019_

## Referee Comment (RC1) · Anonymous Referee #1 · 11 Sep 2019

This is a very timely paper given the widespread use of the VBS combined with the lack of auto-oxidation chemistry in most chemical transport models and box models. Guidance for how to treat auto-oxidation in the VBS is needed. The temperature dependent aspects (for RO2 dimerization and auto-oxidation rate constants) are particularly novel and interesting. Overall, the work is useful, but could be better tied to other studies to determine more concrete parameters and steps forward for the community.

Major comments:

1. Where does the gas-phase chemistry leave and the VBS take over? Page 2 highlights how auto-oxidation is similar to other RO2 fates (+NO or +HO2) whose products can be treated by a VBS. The RO2+HO2, etc is not predicted by the VBS, but a gas-phase mechanism. Would it be more accurate (e.g. to state in the abstract) that "a new

implementation of the VBS that explicitly resolves peroxy radical (RO2) PRODUCTS formed via auto-oxidation"? When do the kernels take over? At a specific volatility? Would a chemical transport model use the VBS (or kernels) to predict auto-oxidation?

2. The work is highly idealized and hypothetical and could be informed by recent mechanistic work on the a-pinene system. Future work (aimed at, for example, estimating auto-oxidation rate constants and RO2+RO2 dimer rates) could also be better informed if some of the already published parameters were considered in this work.

a. The authors assert OH radical-derived RO2 do no auto-oxidize as readily as ozone-initiated ones with a reference to Ehn et al. 2014 (page 4). Berndt et al. 2016 found that quantification of the OH-initiated HOMs from a-pinene are highly sensitive to the detection technique and that previous work likely underestimated them. Revise in consideration of this more recent work.

b. Rather than specifying one value per oxidant for the fraction of RO2 able to undergo auto-oxidation (alpha, page 6), can the authors provide insight into a plausible range? For example, the alpha for OH is set at 0.1 while Vereecken et al. 2007, Berndt et al. 2016, and Pye et al. 2019 all suggest values on the order of 0.2. Can one of those values (0.1 or 0.2) be ruled out or are they both plausible?

c. You may want to consider examining faster auto-oxidation rate constants (3-10 1/s) and add a-pinene RO2+RO2 specific values (Section 3.1.3-3.1.5) based on literature (Zhao et al. 2019).

d. How does the value of gamma (page 8) predicted in this work match up with gamma from mechanistic work (e.g. Zhao et al. 2019)?

e. Why wasn't an actual a-pinene experiment simulated? How well would the parameterization perform?

f. Page 12, line 33: "NOx suppresses dimer yields more aggressively than it does HOM monomer yields." How likely is this to be generally applicable to the atmosphere? Can

you modify rate constants and branching to state with more/less confidence what the NOx modulation of HOM is?

g. Page 13, line 23: what bounds do you propose for the auto-oxidation rate coefficient? The bounds on the carbon yield in Fig 9 can be quite large (the min/max bound is also hard to see and link with monomer vs dimer).

3. How generalizable are the parameters within the a-pinene system and to other systems? Previous work (e.g., Kurten et al., 2017) has highlighted how structure-specific product distributions can be. Can the a-pinene system be treated with a series of general kernels or will we eventually need an explicit mechanism that replaces the specified values of alpha, the dimer kernel, etc. with individual species and unique yields? How important is it to capture the diversity of barriers and rate constants (page 6)? Can you provide some bounds for what might be good enough?

Minor comments:

1. Page 2, line 9-add units after 300

2. Page 2, clarify Co is volatility at temperature, T?

3. Page 6, section 3.1.2: what are the parameters (A) based on?

4. Section 3.1.3 and 3.1.5 both cover RO2 cross reactions. How are they connected?

5. Page 9, line 9-add units after 300. Note additional appearances of 300 should be 300 K throughout the manuscript.

6. Page 11, near line 20, Does the LVOC characterization apply based on the C* at Tref or C* at T? Is the definition of LVOC, ELVOC, etc environment dependent or unique to the species? Also page 11, line 32 "…the volatility classes shift toward higher Co (300) at lower temperature…" is a bit confusing.

References:

Berndt et al. 2016: Hydroxyl radical-induced formation of highly oxidized organic compounds, https://doi.org/10.1038/ncomms13677

Kurten et al. 2017: Alkoxy Radical Bond Scissions Explain the Anomalously Low Secondary Organic Aerosol and Organonitrate Yields From $\alpha$-Pinene + NO3, https://doi.org/10.1021/acs.jpclett.7b01038

Pye et al. 2019: Anthropogenic enhancements to production of highly oxygenated molecules from autoxidation, https://doi.org/10.1073/pnas.1810774116

Vereecken et al. 2007: Low-volatility poly-oxygenates in the OH-initiated atmospheric oxidation of $\alpha$-pinene: impact of non-traditional peroxyl radical chemistry, https://doi.org/10.1039/B708023A

Zhao et al. 2019: Quantitative constraints on autoxidation and dimer formation from direct probing of monoterpene-derived peroxy radical chemistry, https://doi.org/10.1073/pnas.1812147115

---

## Referee Comment (RC2) · Anonymous Referee #2 · 20 Sep 2019

This paper makes a timely and well thought out extension to the 2D-VBS volatility formulation by considering the chemistry of peroxy radicals (RO2) explicitly. The kinetics of autooxidation are specifically included, since these reactions are now thought to lead to small yields of condensible products within a single generation.

The methodology is to run a relatively simple 0-D box model with parameterization of various reaction rates and product yields to obtain the estimated volatility and O:C ratios of the products. The model addresses a range of temperatures and NOx levels. I like the concept of relating the probability of dimer formation to the volatility of the peroxy radical(s) involved in its formation. This is something that should be explored further.

Alpha pinene is used as the test/surrogate molecule in the model. This is appropriate

in the sense that the most is known about alpha pinene. Notwithstanding, there is still a lot we don't know about that molecule, and we know even less about others. The authors are obviously aware of this, and discuss it, but it should probably be made more clear that this is something of an exploratory sensitivity test, rather than an attempt to explicitly represent this chemistry in models at the moment. However, it is a very valuable exercise, and could be modified relatively easily as more data become available.

The paper is logically written and well explained, and can by published subject to the relatively minor comments below.

Technical comments Page 8, line 17. "here we shall explore the possibility that..." I was expecting more exploration, for example with different scenarios. Instead it seems like just one situation was considered.

Page 8 line 26. Is the value of Co(ref) ever defined? Were different values considered (explore!)?

Page 9 line 3. Here (and in a few other places) a-pinene "oxidation" is referred to. Maybe be a little more clear by specifying "ozonolysis".

Page 11, line 5. I don't see the factor of 40-80 in the figure. Looks more like 10 or 20.

Page 11, line 6. Is this the first reference to photolysis being "on"? Maybe it should be mentioned in the general description of the set up on Page 10. Until then, I had assumed the ozonolysis would take place in the dark.

Figure 4 could maybe be made a little clearer. It took me a while, but I eventually figured out that the labels on the right axis corresponded to the HO2 and RO2 for the different NOx levels.

Supplememtal Table, page 2. The branching from AP + O3 seems to contradict the text. Here, the radical RO2 is allowed to isomerize, so it actually corresponds to OxoRO2 in the text. So the yield should be 0.25 not 0.75? The less reactive radical SVOC should

then correspond to RO2 in the text.

Do the SVOC/RO2 radicals participate in subsequent chemistry (reaction with HO2, cross reactions)?

Also, the coefficients in the Table are all 0.75/0.25, while in the text it is stated that alpha(OH) is 0.1 and alpha(NO3) is 0. Just typos? This all needs to be tidied up.

Minor comments Page 1, line 21. "are" is repeated.

Page 12, line 23. "at" should be "a".

Figure 2. Left axis should read "concentration". Caption line 2. Is a little simplistic. Of course the radicals are reacting away all the time. It's just that the source (a-pinene + O3) is reduced)

Figure 4, caption. "before gradually decaying". Does this refer to their behavior with time, or as a function of NO?

Figure 5. This is probably a stylistic thing. I find the lengthy caption inappropriate. Much of this is discussion, which might be better off in the text. I prefer captions to be punchy, with just enough description to be able to understand the figure (which isn't always the case here).

Figure 6, caption. Delete "without".

Figures in general. A couple of times, the top of a curve is missing (figs 3 and 8, for example). Can these be scaled differently, without introducing an extra decade in the Y-axis?

---

## Author Comment (AC1) · 31 Oct 2019

Reviewer 1 Responses

We would like to thank the reviewer for the constructive comments and the helpful suggestions to improve this manuscript, especially in the context of trying to use this work to inform future research in the area, as this aligns very well with our future goals. In the following the reviewer comments are shown in blue, and our responses in black.

1. Where does the gas-phase chemistry leave and the VBS take over? Page 2 highlights how auto-oxidation is similar to other RO2 fates (+NO or +HO2) whose products

can be treated by a VBS. The RO2+HO2, etc is not predicted by the VBS, but a gas-phase mechanism. Would it be more accurate (e.g. to state in the abstract) that "a new implementation of the VBS that explicitly resolves peroxy radical (RO2) PRODUCTS formed via auto-oxidation"? When do the kernels take over? At a specific volatility? Would a chemical transport model use the VBS (or kernels) to predict auto-oxidation?

We added the suggested language to the abstract to clarify that yes, a gas-phase mechanism predicts the reaction yields and the volatility yields are resolved using kernels following that. In this framework the $RO_2$ radicals are always treated within the chemical mechanism and their molecular products resolve into the VBS (so auto-oxidation is treated within the mechanism). The attribute for any $RO_2$ in the chemical mechanism, including within a CTM, to be resolved into the VBS is a volatility ($C°$) estimate and O:C. In this implementation all $RO_2$ are resolved into the VBS but this is in part because we are only modeling $\alpha$-pinene oxidation and even the initial $RO_2$ has a relatively low (SVOC) $C°$; implementations for a richer chemistry will be free to select any subset of all $RO_2$ to resolve into a VBS and also to lump $RO_2$ were that desired.

2. The work is highly idealized and hypothetical and could be informed by recent mechanistic work on the a-pinene system. Future work (aimed at, for example, estimating auto-oxidation rate constants and RO2+RO2 dimer rates) could also be better informed if some of the already published parameters were considered in this work.

We chose not to directly simulate individual experiments because we are reluctant to attach meaning to the tuned parameters that would result in such an exercise. Further, it would require a well-vetted model of each experiment, which would in our opinion obscure the message of this paper. This is based on past experience. As an example, equilibrium interpretation of chamber experiments, extending from Odum et al., through many of our own VBS analyses of chamber experiments, was ultimately found to be erroneous for two reasons. First, the dynamics of SOA condensation is often important, and, second, vapor losses to chamber walls can vary between experiments

and within experiments in ways that make mass yields appear low at low overall aerosol mass (really Fuchs-adjusted surface area) loadings. The only parameters in the equilibrium model capable of reproducing the rising mass yields with rising loadings were the VBS bin mass yields themselves. The resulting fits were not only good but they were highly predictive (including temperature effects) in part because they implicitly contained chamber effects and these effects reasonably transferred from one experiment to another. However, the entire tongue of (UE)LVOCs that are the major topic of this paper was completely missing from those fits, and the inference was an SOA population that was far more "semi volatile" than now appears to be appropriate. This was compounded by the lack of a chemical mechanism to explain that low-volatility tongue (without auto oxidation, HOMs, and dimers). The more semi-volatile SOA was far more consistent with the then canonical oxidation mechanisms, making the whole scheme self consistent but, ultimately, incomplete in important ways.

Rather than directly comparing to experiments, our approach is to use the (often uncertain and sometimes contradictory) current experimental findings as constraints on the kinetic parameters of our model current simple model. There are at least two areas where our simple scheme may well ultimately require more nuance before more direct comparisons with experiments is fruitful. Both are related to the diversity of chemical behavior likely in the (many) peroxy radicals arising even from a single precursor such as $\alpha$-pinene. Fortunately, the rate coefficients and products for reactions with NO and $HO_2$ appear to be relatively constant across a sequence of $RO_2$ radicals, but the $RO_2$ cross reactions and also the auto-oxidation rate constants are highly variable. Rather than a single tongue of progressively more oxidized $RO_2$, which we currently employ, it is likely that we will require at least a "less reactive" and "more reactive" $RO_2$ at each stage of auto-oxidation. It is unclear whether "less" and "more" will apply equally to the internal H-atom transfers and the $RO_2$ cross reactions, so even more diversity may be in order.

It is unlikely that aggregate fits and comparisons with bulk measures such as SOA

mass and nucleation rates will be useful constraints for these $RO_2$ kinetics parameters. Rather, the emerging set of kinetics data on individual reactions will need to be used to define the mechanism itself; however, in our opinion the phase space of $RO_2$ reactions is too large to make that fruitful right now.

a. The authors assert OH radical-derived RO2 do no auto-oxidize as readily as ozone initiated ones with a reference to Ehn et al. 2014 (page 4). Berndt et al. 2016 found that quantification of the OH-initiated HOMs from a-pinene are highly sensitive to the detection technique and that previous work likely underestimated them. Revise in consideration of this more recent work.

This is a valid point but there are other constraints as well. For example, Kirkby et al. (2016) found that the nucleation rate associated with additional oxidation of $\alpha$-pinene by OH (after initial ozonolysis, as simulated here) had only a small effect on the observed new-particle formation rates. This is consistent with OH oxidation having lower overall HOM yields than ozonolysis (and inconsistent with similar yields). However, the point is well taken that it is appropriate to consider a range of the HOM yields from OH oxidation, and so we will revise the manuscript to consider a higher HOM yield from OH in addition to our base-case lower yield. This does not change our conclusions about the temperature and NO dependence of the processes.

b. Rather than specifying one value per oxidant for the fraction of RO2 able to undergo auto-oxidation (alpha, page 6), can the authors provide insight into a plausible range? For example, the alpha for OH is set at 0.1 while Vereecken et al., 2007, Berndt et al., 2016, and Pye et al., 2019 all suggest values on the order of 0.2. Can one of those values (0.1 or 0.2) be ruled out or are they both plausible?

An $\alpha$ value of 0.2 is completely reasonable within this framework. We do not present a full sensitivity analysis but instead hold some parameters constant, such as $\alpha$. However, we have explored the sensitivity parameters including $\alpha$, the $RO_2$ isomerization barrier height and pre-factor, the dimer formation rate constant, etc. What we find is a substantial co-variance among these parameters suggesting a complex parameter phase space with many possible parameter sets producing equally "good" results. Again, because our objective in this work is to explore the sensitivity of the HOM yields to temperature and NO, more than to describe their absolute yields, we chose to hold this parameter fixed for the sake of simplicity. The effect of these two parameters on the temperature and $NO_x$ dependencies are relatively small, as can be seen in the figure at the end of the response.

c. You may want to consider examining faster auto-oxidation rate constants (3-10 1/s) and add a-pinene RO2+RO2 specific values (Section 3.1.3-3.1.5) based on literature (Zhao et al. 2019).

We do address this by allowing the isomerization pre-factor to vary, as shown in Figure 9. Qualitatively, the high-temperature HOM (monomer + dimer) yields increase to essentially all of the HOMs allowed by the parameter $\alpha$, and only drop once it is cold enough to slow down the isomerization enough for bimolecular reactions to compete. Likewise, it requires proportionally more NO to quench the $RO_2$. As we discussed above, it is possible that to adequately describe auto-oxidation, its effects on new-particle formation as well as SOA formation, and the sensitivity of both to NO, T, $HO_2$, etc, we will need to treat "fast" and "slow" reacting $RO_2$. There is ample evidence in the kinetic literature for a high degree of variability in both the $RO_2$ cross reactions and the auto-oxidation rate. Equally, we are unsure whether the $RO_2$ species are directly comparable in the relatively low-concentration, long timescale experiments such as CLOUD and the relatively high-concentration, short timescale experiments such as flowtubes.

We will add text in a revised manuscript to make this clearer.

d. How does the value of gamma (page 8) predicted in this work match up with gamma

from mechanistic work (e.g. Zhao et al. 2019)?

The $\gamma$ value ranges from $10^{-8}$ for the least oxidized RO$_2$s to 0.9 for the most oxidized at 298 K. The value of gamma is dependent on the $Co_{ref}, which is dependent on temperature, but these values are the lowest for each peroxy radical over the temperature range inve$ values agree with their work. However, their work also predicts slower rate constants than used here, which had we used, we would need higher gamma values.

e. Why wasn't an actual a-pinene experiment simulated? How well would the parameterization perform?

As discussed above, we did not simulate individual experiments because the large number of tunable parameters would allow us to reproduce a data set fairly easily without providing more clarity on the parameters. For example making the auto-oxidation rate constant faster would cause more oxidized products to be form and keeping everything else the same would lead to more HOM formation. But if we simultaneously slow down the dimerization rate constants and the unimolecular rate constant, we may be able to return to the same picture we present in this paper with a different set of parameters. We would also need to develop a corresponding chamber or flow-tube model, with appropriate wall-loss treatment, to properly model any individual experiment. The uncertain parameters associated with that reactor model would further confound the analysis.

f. Page 12, line 33: "NOx suppresses dimer yields more aggressively than it does HOM monomer yields.? How likely is this to be generally applicable to the atmosphere? Can you modify rate constants and branching to state with more/less confidence what the NOx modulation of HOM is?

Lehtipalo et al., 2018 reported strong decreases in dimer yields coupled with an increase in total HOM yields due to nitrate-containing monomers in chamber experiments. However, this is another case where the full potential array of parameters leaves several possibilities open. One possibility is that the isomerization rate coefficient important here remain relatively constant as the auto-oxidation progresses (this is our base case). Another is that the initial isomerization rate coefficient is relatively slow and these accelerate with increasing functionalization. In that case the initial $RO_2$ would be a significant bottleneck and NO could substantially quench both HOM monomer and dimer formation.

g. Page 13, line 23: what bounds do you propose for the auto-oxidation rate coefficient? The bounds on the carbon yield in Fig 9 can be quite large (the min/max bound is also hard to see and link with monomer vs dimer).

We allow the rate coefficient to vary through two parameters – the barrier and the A factor (Fig 8 and Fig 9). We suggest bounds of $7500 < E_a < 9000$ K or approximately 15-18 kcal/mol for the barrier (with the rate constant fixed at 300) and $(10, 7, 6) \times 10^{7\pm1}$ This low end range allows auto-oxidation to be competitive with fast bimolecular reactions

and to have the strong temperature dependence that is seen experimentally. The high end of this range is when the barrier becomes too high to overcome even at high temperatures where we expect to see a significant fraction of auto-oxidized products.

3. How generalizable are the parameters within the a-pinene system and to other systems? Previous work (e.g., Kurten et al., 2017) has highlighted how structure specific product distributions can be. Can the a-pinene system be treated with a series of general kernels or will we eventually need an explicit mechanism that replaces the specified values of alpha, the dimer kernel, etc. with individual species and unique yields? How important is it to capture the diversity of barriers and rate constants (page 6)? Can you provide some bounds for what might be good enough?

It is important to capture the diversity of of barriers and rate constants, however being able to group peroxy radicals as we did here would allow for a much simpler implementation into chemical models. Without specific rate constants for every peroxy radical being produced, here we seek to group the peroxy radicals by the extent of oxidation they have undergone. There is evidence in Kurten et al and Zhao et al 2019 that these trends may be present, however in the future comparison with a more explicit chemical mechanism would be able to tell us what is "good enough."

1. Page 2, line 9-add units after 300

Done.

2. Page 2, clarify Co is volatility at temperature, T?

We plot everything using C*(300) as the x-axis so as to visualize how the products of the chemistry are changing rather and inferring how this would affect condensation. This language was added on page 2.

3. Page 6, section 3.1.2: what are the parameters (A) based on?

The value of A was chosen to give auto-oxidation rate constants of about 0.01 s$^{-1}$ while maintaining a high barrier for a strong temperature dependence.

4. Section 3.1.3 and 3.1.5 both cover RO2 cross reactions. How are they connected?

They are the same reaction, however not all association reactions form dimers as some may form 2 alkoxy radicals that stabilize with other radicals to form monomers. Section 3.1.3 and 3.1.4 were switched as 3.1.3 and 3.1.5 are in fact very closely related.

5. Page 9, line 9-add units after 300. Note additional appearances of 300 should be 300 K throughout the manuscript.

This was fixed.

6. Page 11, near line 20, Does the LVOC characterization apply based on the C* at Tref or C* at T? Is the definition of LVOC, ELVOC, etc environment dependent or unique to the species? Also page 11, line 32 ?. . .the volatility classes shift toward higher Co (300) at lower temperature. . .? is a bit confusing.

The LVOC range is different at every temperature. It is visually represented in Figs 5 and 6, but we agree the written language is confusing and page 11 line 20 has been restated.

[Figure]

**Fig. 1.** The NOx dependence of monomer (solid lines) and dimer yields (dashed lines) at alpha_OH = 0.2 (dark colors) and alpha_OH = 0.2 (light colors).

[Figure]

**Fig. 2.** The temperature dependence of monomer (solid lines) and dimer yields (dashed lines) at alpha_OH = 0.2 (dark colors) and alpha_OH = 0.2 (light colors).

---

## Author Comment (AC2) · 31 Oct 2019

Reviewer 2 Responses

We would like to thank the reviewer for the constructive comments and for bringing to our attention some clarity issues that we have attempted to address. In the following the reviewer comments are shown in blue, and our responses in black.

Page 8, line 17. "here we shall explore the possibility that. . ." I was expecting more exploration, for example with different scenarios. Instead it seems like just one situation was considered.

[Figure]

The efficiency of dimer formation is in our assessment a rapidly moving target. Other than the empirical observation that very highly oxidized $RO_2$ appear to make dimers with high efficiency and very small $RO_2$ (i.e. $CH_3O_2$) appear to make little or no dimer, the territory in the middle is uncertain. The efficiency of the avoid curve crossing may well scale with some combination of the oxygenated functional groups near the ROO moieties as well as cluster stabilization allowing for a longer interaction time (functionally the same phenomenon we are exploring here). For this reason we kept a single mechanism for dimer formation in this work; however, it is important to note that this causes the dimer production in our model to extend to ever less functionalized, less oxidized $RO_2$ as temperature decreases. Temperature dependent measurements of dimer yields with instruments sensitive to the full range of dimers would provide an excellent constraint here.

Page 8 line 26. Is the value of Co(ref) ever defined? Were different values considered (explore!)?

The value of C°(ref) used in this work is $10^{-2}$ at 298 K and moves 1 order of magnitude lower in volatility per 10 K reduction in temperature. We explored different values of C°(ref) during model development; however, we chose to hold it constant at each temperature for the results presented here to limit the number of tunable parameters is the simulation.

Page 9 line 3. Here (and in a few other places) a-pinene "oxidation" is referred to. Maybe be a little more clear by specifying "ozonolysis".

We use the word "oxidation" to add generality as $\alpha$-pinene may be oxidized by ozone, OH, or $NO_3$ when $NO_x$ is present. Our simulations are thus driven initially by ozonolysis but include all three oxidants.

[Figure]

Page 11, line 5. I don't see the factor of 40-80 in the figure. Looks more like 10 or 20.

The $RO_2$ in Fig. 4 are the "$OxRO_2$" including at least one -OOH group as shown also in Fig. 3. As one can see in Fig. 2, the "simple" $RO_2$ also has a maximum concentration above $2.5 \times 10^8$ cm$^{-3}$ so the sum is well over $2.5 \times 10^8$. We have clarified this in the figures and text.

Page 11, line 6. Is this the first reference to photolysis being "on"? Maybe it should be mentioned in the general description of the set up on Page 10. Until then, I had assumed the ozonolysis would take place in the dark.

In the revised manuscript we make it clear at the onset that photochemistry is involved.

Figure 4 could maybe be made a little clearer. It took me a while, but I eventually figured out that the labels on the right axis corresponded to the HO2 and RO2 for the different NOx levels.

The top legend incorrectly labeled the blue curves as "$NO_x$"; they are "NO" and we have corrected this. We have also added to the caption to emphasize that the labels along the right-hand y-axis refer to the $NO_x$ concentration for each simulation by inserting "The numbers along the right-hand y-axis refer to the $NO_x$ concentration for each simulation, which is also indicated by the shading of each curve, going from light at low $NO_x$ to dark at high $NO_x$".

Supplemental Table, page 2. The branching from AP + O3 seems to contradict the text. Here, the radical RO2 is allowed to isomerize, so it actually corresponds to OxoRO2 in the text. So the yield should be 0.25 not 0.75? The less reactive radical SVOC should then correspond to RO2 in the text.

Kudos and thank you for close reading! This was a typo in the supplemental material, which we corrected. The main text is correct.

Do the SVOC/RO2 radicals participate in subsequent chemistry (reaction with HO2, cross reactions)?

Yes, the peroxy radicals that we do not allow to isomerize may still participate in any of the termination chemistry.

Also, the coefficients in the Table are all 0.75/0.25, while in the text it is stated that alpha(OH) is 0.1 and alpha(NO3) is 0. Just typos? This all needs to be tidied up.

Once again, thank you. These were typos in the supplemental table; the main text is correct and we corrected the typos.

Page 1, line 21. "are" is repeated.

Fixed.

Page 12, line 23. "at" should be "a".

Yep.

Figure 2. Left axis should read "concentration". Caption line 2. Is a little simplistic. Of course the radicals are reacting away all the time. It's just that the source (a-pinene + O3) is reduced)

The revised caption reads "as $\alpha$-pinene decays and the $RO_2$ react away"

Figure 4, caption. "before gradually decaying". Does this refer to their behavior with time, or as a function of NO?

We added "As a function of time" to the end of the caption to make this clear.

Figure 5. This is probably a stylistic thing. I find the lengthy caption inappropriate. Much of this is discussion, which might be better off in the text. I prefer captions to be punchy, with just enough description to be able to understand the figure (which isn't always the case here).

We simplified the caption and moved some discussion to the main text while retaining enough substance so that a casual reader can understand the figure while scanning over just the "storyboard" of figures, which is our objective. We also reworked the caption so that the sense is temperature increasing; this allows the reader to more easily scan the figures in the natural direction from left to right.

Figure 6, caption. Delete "without".

Done.

Figures in general. A couple of times, the top of a curve is missing (figs 3 and 8, for example). Can these be scaled differently, without introducing an extra decade in the Y-axis?

We are focusing on the oxidized $RO_2$ ($Ox_nRO_2$) but the text and captions were not clear. We have revised them to make this clearer.